# A Novel Sensor Fusion Approach for Precise Hand Tracking in Virtual Reality-Based Human—Computer Interaction

**DOI:** 10.3390/biomimetics8030326

**Published:** 2023-07-22

**Authors:** Yu Lei, Yi Deng, Lin Dong, Xiaohui Li, Xiangnan Li, Zhi Su

**Affiliations:** 1College of Humanities and Arts, Hunan International Economics University, Changsha 410012, China; xiaoyu9767@gmail.com; 2College of Physical Education, Hunan International Economics University, Changsha 410012, China; 3Institute of Sports Artificial Intelligence, Capital University of Physical Education and Sports, Beijing 100091, China; 4Department of Wushu and China, Songshan Shaolin Wushu College, Zhengzhou 452470, China; 5Department of History and Pakistan, University of the Punjab, Lahore 54000, Pakistan; 6Yantai Science and Technology Innovation Promotion Center, Yantai 264005, China; 7Department of Information, School of Design and Art, Changsha University of Science and Technology, Changsha 410076, China; 006034@csust.edu.cn

**Keywords:** sensor fusion, hand tracking, human–computer interaction, neural networks, multi-modal data

## Abstract

The rapidly evolving field of Virtual Reality (VR)-based Human–Computer Interaction (HCI) presents a significant demand for robust and accurate hand tracking solutions. Current technologies, predominantly based on single-sensing modalities, fall short in providing comprehensive information capture due to susceptibility to occlusions and environmental factors. In this paper, we introduce a novel sensor fusion approach combined with a Long Short-Term Memory (LSTM)-based algorithm for enhanced hand tracking in VR-based HCI. Our system employs six Leap Motion controllers, two RealSense depth cameras, and two Myo armbands to yield a multi-modal data capture. This rich data set is then processed using LSTM, ensuring the accurate real-time tracking of complex hand movements. The proposed system provides a powerful tool for intuitive and immersive interactions in VR environments.

## 1. Introduction

Over the past few decades, human–computer interaction (HCI) has undergone significant evolution, fundamentally transforming the way individuals engage with technology and influencing a wide range of domains, including education, gaming, entertainment, and healthcare [1,2,3,4]. Concurrently, virtual reality (VR) has witnessed substantial progress, encompassing advancements in high-resolution displays, enhanced tracking systems, realistic haptic feedback, and the emergence of stand-alone VR headsets [5,6,7,8]. Of particular interest is VR-based HCI, which has attracted considerable attention due to its immersive nature and its potential to enrich user experiences across diverse fields [9]. Within VR-based HCI, the tracking of multiple body parts, including the head, eyes, and hands, is a common practice [10,11,12]. Hand tracking, in particular, plays a critical role as it enables users to interact naturally and intuitively with the virtual environment through gestures, actions, and manipulations.

However, existing hand-tracking technologies that rely on a single sensing modality exhibit limitations [13]. Optical-based methods may be prone to occlusions and lighting sensitivity, while depth sensors can be affected by environmental factors and may lack the required resolution for accurate tracking [14,15]. EMG devices, although capturing muscle activity, may not capture fine motor skills or subtle gestures precisely [16]. Additionally, relying solely on data from a single sensing modality may lead to instability and reduced accuracy in hand tracking [17], which may significantly impact the effectiveness and usability of hand-tracking technologies in HCI applications. To overcome these challenges, it is crucial to explore novel approaches that integrate multiple sensing modalities to enhance perception capabilities and ensure stable data sources for precise hand tracking in VR-based HCI.

In light of these challenges, we present a novel sensor fusion scheme for precise hand tracking in VR-based HCI applications. Our approach integrates the latest VR headset, PICO 4, with a range of sensors, including six Leap Motion controllers, two RealSense depth cameras, and two Myo armbands [18]. Figure 1 illustrates the potential application scenarios of our system, such as rehabilitation training and medical skills training. By leveraging these diverse sensors, our multi-modal hand-tracking acquisition system captures a comprehensive and complementary set of information sources, enabling more natural and intuitive human–computer interaction. Through the fusion of data from these different sensors, our approach effectively overcomes the limitations inherent in single-modality systems, providing a robust and accurate hand-tracking solution in various VR environments.

The proposed approach offers several key contributions:(1)This paper introduces a multi-modal hand-tracking acquisition system that achieves enhanced tracking accuracy and reliability across diverse scenarios.(2)The proposed method ensures real-time performance, enabling seamless integration with existing VR applications.(3)By facilitating more natural and immersive interactions, this approach opens up possibilities for advancements in VR-based HCI across various domains, such as gaming, education, and healthcare.

These contributions collectively contribute to the advancement and broader adoption of VR technology in human–computer interaction.

This article is structured as follows: First, a comprehensive literature review is presented, examining the current landscape of hand-tracking technologies and sensor fusion methods in VR-based HCI. Next, the methodology and system architecture of our multi-modal hand-tracking acquisition system are described in detail, including the sensor fusion approach and the employed algorithms. Subsequently, experimental results and performance analyses are reported, highlighting the strengths of our proposed method in comparison to existing techniques. Finally, conclusions are drawn, and potential directions for future research in this field are discussed.

## 2. Related Work

In this section, we provide a comprehensive review of the current state-of-the-art hand-tracking technologies in the context of VR-based HCI. We particularly highlight the limitations and challenges faced by single-sensor methods. Moreover, we make a table to compare various hand-tracking systems, as shown in Table 1. Through an analysis of previous research in this field, our aim is to identify the gaps in existing approaches and emphasize the significance of advanced and precise methods, particularly those that incorporate sensor fusion techniques.

To propose a novel approach for precise hand tracking, it is essential to have a comprehensive understanding of the evolution of hand tracking technologies. During the 1960s and 1970s, capacitive sensors were the primary method for hand tracking [19], but they suffered from issues such as low accuracy, poor reliability, and high cost. In the 1980s and 1990s, wired glove-based input devices, such as the DataGlove and the CyberGlove, were widely used [20]. They incorporated embedded sensors in the gloves to detect hand movements and gestures but turned out to be very cumbersome. Technological advancements led to the development of more accurate techniques, such as the utilization of accelerometers [21], infrared cameras [22], and fiber-optic bend sensors [23]. Such advancements resulted in improved precision for glove-based control interfaces. Furthermore, the introduction of wireless and easy-to-wear gloves further enhanced the usability of these systems. Subsequently, vision-based hand-tracking methods emerged with the advent of digital cameras and computer vision algorithms [24,25] that utilized cameras to capture hand images and employed various image processing techniques, such as skin color segmentation and contour extraction, to track hand positions and movements. Vision-based methods offered contactless tracking, providing a more natural and comfortable user experience compared to non-vision-based recognition methods like data gloves or electromagnetic waves [26]. However, they were often sensitive to lighting conditions and susceptible to occlusions.

In the late 2000s and early 2010s, depth cameras like Microsoft’s Kinect and Intel’s RealSense have played a significant role in advancing hand-tracking technology [27]. These depth-based methods leverage depth information to address some limitations of vision-based approaches, such as sensitivity to lighting conditions and occlusions. However, challenges still existed in terms of tracking accuracy, especially with complex hand poses and self-occlusions [28]. To overcome the limitations of previous methods, researchers began applying machine learning techniques in the mid-2010s, specifically convolutional neural networks (CNNs) and other deep learning models, to hand tracking [29,30]. These approaches demonstrated higher tracking accuracy and robustness compared to earlier methods. Notably, the availability of large-scale hand pose datasets and the utilization of data augmentation techniques have played a crucial role in the success of machine learning-based hand tracking methods [31,32].

Despite the advancements in machine learning-based hand tracking methods that have improved tracking accuracy, the utilization of a single sensor for hand tracking still imposes several limitations, including data instability [33,34,35], limited field of view [36], low tracking accuracy [13], and sensor-specific characteristics [37]. To overcome the above-mentioned limitations, sensor fusion approaches have been proposed to integrate multiple sensors and enhance the effectiveness of hand tracking methodologies. For instance, the fusion of Magnetic/Angular-Rate/Gravity (MARG) sensors and IMUs offers the potential to compensate for individual sensor limitations and achieve precise hand tracking across a broad range of motions [38]. However, challenges arise when the sensor module experiences significant accelerations or rotations, which can adversely affect the performance of the tracking algorithm and result in suboptimal accuracy. Furthermore, the fusion of electromyography and depth sensors allows for the accurate tracking of arm and hand poses, thereby supporting supplementary applications such as held object recognition and environment mapping [39]. Nevertheless, accurately estimating hand poses in the presence of significant self-occlusion remains a highly complex challenge.

In the domain of VR-based HCI, hand tracking holds significant importance due to its potential to enable more natural and intuitive interactions between users and virtual environments [40,41]. Hand tracking in VR-based HCI finds numerous critical applications, showcasing its crucial role in shaping immersive experiences. Gesture recognition, a fundamental aspect of hand tracking, allows users to communicate with virtual environments through a variety of hand gestures. In the study by Liliana et al. [42], a pipeline combining movement sensors, a binary image representation of a gesture shape, and a density-based CNN was proposed. This pipeline, combined with HTC Vive, Kinect, and Leap Motion, has achieved accurate hand gesture recognition with a remarkable 97.7% accuracy. Additionally, hand tracking enables the design of innovative control interfaces for VR-based HCI systems [43]. In the study by Song et al. [44], an improved hand gesture control interface called GaFinC was developed for 3D modeling manipulation tasks. GaFinC reduced user physical fatigue while maintaining high intuitiveness, providing users with greater freedom and flexibility when interacting with virtual environments. In the study by Ovur et al. [45], an adaptive multisensor fusion methodology is proposed for hand pose estimation with two Leap Motions, which can perform stable and steady hand pose estimation in real-time, even when a single sensor is unable to detect the hand, improving the smoothness of pose estimations without being affected by occlusion on one sensor. In the study by Rigas et al. [46], a hybrid eye-tracking technique that fuses photosensor and video oculography to provide robustness to sensor shifts for high-speed eye tracking is proposed and evaluated for use in emerging head-mounted devices, which is exceptionally enlightening.

The field of VR-based HCI has witnessed advancements in various applications and technologies, including sensor algorithms and user-independent gesture classification methods [47,48,49]. In a study by Butt et al. [50], an adaptive covariance-based Extended Kalman Filter (EKF) algorithm was proposed for sensor fusion in ambulatory motion capture and gait analysis using wearable MEMS-based MIMUs. The algorithm effectively addressed challenges related to degraded performance due to prolonged use and inhomogeneous magnetic fields. It featured gyro bias updates and accurate orientation estimation, along with a novel correction term to mitigate drift in individual joint angles. Alfaro and Trejos [51] presented a user-independent gesture classification method that combines EMG and IMU data through sensor fusion. This approach enables enhanced control of wearable mechatronic devices during robot-assisted therapies, achieving impressive classification accuracies of up to 92.9%. Furthermore, advancements in VR-based HCI can be observed in intelligent vehicle applications. In a study by Vu et al. [52], a novel algorithm for object detection, tracking, and classification was introduced. The algorithm leveraged multiple sensors and employed raw sensor data processing and fusion techniques. It demonstrated improved reliability and accuracy in real-life traffic testing scenarios. To enhance tracking accuracy in VR-based HCI, Bazo et al. [53] proposed a system that integrates radiofrequency-based positioning with computer vision-based human pose estimation techniques. By fusing radiofrequency sensor identities with unidentified body poses and estimated body parts, the system achieved a substantial reduction in positioning errors of nearly 46%. Advancements in sensor algorithms, sensor fusion, and gesture classification methods underscore the progress in VR-based HCI, offering promising avenues for more precise and immersive user experiences across diverse domains.

Previous investigations have primarily focused on single-sensor or two-sensor approaches, exhibiting inherent constraints such as data instability, susceptibility to occlusions, and diminished tracking accuracy. To surmount these challenges, we propose an innovative methodology that synergistically integrates four types of sensors: Kinect, RealSense, Leap Motion and Myo armband, to facilitate the meticulous monitoring of intricate hand movements and gestures. By harnessing the complementary attributes of multiple sensors, this approach effectively circumvents the limitations associated with individual sensors, thereby furnishing heightened precision and comprehensive hand-tracking insights.

## 3. Systems Overview

This part details the advancement of each module covered in our system with their characteristics and the embedded systems utilized for their integration.

### 3.1. Leap Motion Controllers

The Leap Motion controller is widely recognized for its sub-millimeter precision, making it a pioneering system for tracking gestures and positions. This surface-independent sensor is specifically designed for immersive stereo 3D interaction systems, providing notable advantages over conventional multi-touch solutions. It enables the natural manipulation of stereoscopically presented objects, offering a more intuitive and immersive experience. Leveraging stereo vision, Leap Motion operates as an optical tracking system, enabling the monitoring and tracking of hands, fingers, and finger-like tools with exceptional accuracy and real-time tracking frame rates. Moreover, the gesture recognition controllers exhibit the capability to capture individual motion parameters within their visual range, which resembles a pyramid with the device’s center as the apex. The dimensions of the controller are 80 mm in length, 30 mm in width, and 113 mm in height. It has a preferred tracking depth range of 10 cm to 60 cm, with a maximum depth capability of 80 cm. The field of view typically spans 140${}^{\circ }$ × 120${}^{\circ }$, and the operating frequency stands at 120 Hz, allowing for the capture of images in a fraction of 1/2000 s. By analyzing changes in displacement, rotation, and scale between consecutive frames, the gesture recognition controller software efficiently extracts comprehensive movement information.

### 3.2. RealSense Depth Cameras

Numerous stereoscopic depth cameras have been developed, often employing similar imagers or imaging processors. However, our approach differs in certain aspects. RealSense depth cameras have been specifically designed to capture the precise and detailed depth information of a scene, with cameras employing active infrared stereo depth-sensing technology to capture accurate depth data. Additionally, RealSense cameras offer additional features, including RGB color imaging, infrared imaging, and seamless compatibility with Intel’s RealSense SDK, providing access to a range of tools and capabilities. The compact depth-aware camera, measuring 90 mm × 25 mm × 25 mm, provides a depth stream output resolution of 1920 × 1080 at 90 fps. With a minimum depth distance (Min-Z) of 28 cm, it can track up to 22 joints in each hand, enabling precise hand and finger tracking. Operating as a stereo-tracking solution, the depth-aware camera leverages the power of stereo vision to enhance depth perception and improve tracking accuracy. The RealSense depth camera has a wide field of view (FOV) of 87${}^{\circ }$ × 58${}^{\circ }$ (+3), capturing depth information across a large area, which is particularly valuable for robotics, augmented reality, and virtual reality applications. It covers a depth range of 0.2 m to 10 m, allowing for reliable depth sensing in various scenarios. Additionally, RealSense cameras can be easily connected to a computer via USB and are compatible with multiple operating systems, including Windows, Linux, and macOS. One of the notable advantages of RealSense cameras is their ability to capture depth information at high frame rates, reaching up to 90 frames per second. This high frame rate enables real-time tracking and enhances responsiveness in interactive applications.

### 3.3. EMG Sensors

The EMG signal control armband offers a distinct approach compared to the somatosensory interaction methods discussed earlier. It serves as a device for somatosensory control by capturing the bioelectric changes in the user’s arm muscles. Currently, the EMG signal control armband is effective in recognizing actions such as fist gestures, arm swinging, and hand and finger spreading. However, due to the weak nature of the muscle signals, precise collection and processing of the EMG signal are crucial for optimal performance. The Myo electric signal control armband comprises eight bioelectric sensor units of various sizes and thicknesses, each equipped with three electrodes. A total of 24 electrodes play a vital role in capturing the bioelectrical variations produced during arm muscle movements. Furthermore, the Myo electric signal control armband incorporates a built-in three-axis accelerometer, a three-axis gyroscope, and an ARM processor akin to that found in mobile phones for data processing. It employs Bluetooth 4.0 for data transmission and features a Micro-USB interface for charging. With a weight of 95 g, the armband ensures the precise detection of subtle changes in muscle activity while minimizing the impact of noise interference.

### 3.4. VR Headset

The PICO 4 VR headset provides users with a truly immersive virtual reality experience, allowing for natural and intuitive interactions in virtual environments. Unlike traditional headsets, the PICO 4 is a standalone device, eliminating the need for a computer or gaming console. It is powered by a Qualcomm Snapdragon 845 processor with 4 GB of RAM and 128 GB of storage, ensuring powerful performance and ample storage capacity. Equipped with 6 degrees of freedom (6DoF) tracking and two 6DoF controllers, the PICO 4 ensures precise and responsive hand tracking in all directions. The headset features a wide field of view (FOV) of 101 degrees, providing users with an expansive visual experience. The 75 Hz refresh rate contributes to a smooth and seamless VR experience, reducing motion sickness and maximizing user comfort during prolonged use. Connectivity is made seamless with Wi-Fi 6 and Bluetooth 5.0 support, allowing for easy integration with other devices. The lightweight design, adjustable head strap, and comfortable fit of the PICO 4 make it ideal for extended VR sessions, ensuring user comfort throughout.

### 3.5. System Framework

The integration of hand-tracking technologies and sensor fusion methods in VR-based HCI has significant implications for various domains, including education, gaming, entertainment, and healthcare. This integration capitalizes on high-resolution displays, advanced tracking systems, realistic haptic feedback, and state-of-the-art standalone VR headsets, resulting in enhanced user experiences. The system framework, as depicted in Figure 2, provides a comprehensive overview of the proposed approach, which facilitates the fusion of multiple sensing modalities to enhance perception capabilities and ensure stable data sources for precise hand tracking in VR-based HCI applications. During the training phase, annotations from different observers, such as researchers, supervisors, experts, and other participants, can be added to enrich the dataset. This inclusion contributes to a more diverse and comprehensive dataset for training and evaluation purposes. A well-structured timeline allows for the playback and analysis of historical data, facilitating research evaluation. Human factors indicators can be obtained through this integration, laying the foundation for a robust and accurate hand-tracking solution in diverse VR environments. By leveraging the capabilities of the PICO 4 VR headset and a combination of six Leap Motion controllers, two RealSense depth cameras, and two Myo armbands, improved tracking accuracy and reliability can be achieved across various scenarios. This study employs a methodology that combines experimental and comparative analysis methods with literature analysis methods. The experimental approach allows for the evaluation of the naturalness and immersiveness of interactions in multiple domains, while the comparative analysis provides insights into the strengths and limitations of different approaches. These findings, along with the literature analysis, form the basis for a conceptual case study, highlighting the potential of the proposed framework in enabling intuitive and immersive interactions in VR-based HCI applications.

### 3.6. Construction of Hand Tracking System

Our hand tracking system comprises two subsystems (Figure 3), one for each hand, which utilize multiple technologies to achieve high-precision tracking of hand and finger movements. The primary motivation behind this design is to maximize the accuracy of hand motion recognition by utilizing sensor fusion techniques and avoiding the possibility of tracking loss due to interference or occlusion.

Each subsystem incorporates three Leap Motion controllers that collaboratively track hand motion, mitigating the risk of tracking loss compared to using a single controller. Leap Motion, as a gesture recognition technology, provides high-resolution tracking capabilities down to the level of individual joints in the fingers. Its tracking capabilities are accurate to 22 minor joints, enabling the detection and monitoring of fine-grained movements.

In addition to the Leap Motion controllers, each subsystem is equipped with a Realsense depth camera to capture depth images of hand activities. This additional perception layer enhances the system’s understanding of the spatial configuration and motion of the hand, further enhancing tracking accuracy.

Our current setup, as shown in Figure 4, predominantly features sensors directed towards the center of the interaction cubes to effectively capture the majority of the hand movements and interactions within this area. We acknowledge the potential issue of self-occlusion where fingers or hand parts may occlude themselves from the sensor’s perspective. To mitigate this, sensors have also been placed at the front, establishing three-dimensional tracking from different angles. A Kalman filtering algorithm is further implemented to reduce occlusion-induced tracking errors, thereby enhancing tracking reliability.

However, we recognize that there could still be self-occlusion instances in some areas due to the lack of sensors on each face of the cubes, which remains a limitation of our current system design. In this study, our primary aim was to develop a tracking setup that offers improved reliability compared to single-tracker systems. We acknowledge that the specific arrangement of sensors significantly impacts the accuracy of tracking, which warrants further exploration. Future work will focus on experimenting with various sensor placements to minimize self-occlusion and improve overall tracking accuracy.

Alongside these vision-based sensors, each subsystem includes a Myo Armband worn on the forearm to collect electromyography (EMG) data. These data reflect the electrical activity of muscles and provides insights into central control factors, characteristics of muscle excitation conduction speed, and fatigue level. This feature is particularly beneficial for users undergoing limb rehabilitation assessments or those unable to wear gloves due to various reasons. The system can measure joint mobility, physiological signals, and EMG signals to provide a comprehensive evaluation of hand function.

The system runs on six computers, each installed with Ubuntu 20 and running the Robot Operating System (ROS) for the seamless integration and management of the various sensors. Data collected by the ROS are then forwarded to an external computer running MATLAB for processing. The processed data are then communicated in real-time with a virtual reality (VR) environment running on a PICO 4 device, developed using Unity. This connection allows for the real-time interaction between the user’s hand movements and the VR environment, creating a fully immersive and responsive experience.

This robust design of our hand tracking system, which combines hand motion capture technology, comprehensive detection technology, EMG signal collection, Realsense depth perception camera, and Leap Motion gesture recognition, provides an accurate and sensitive tool for understanding hand and finger motion across a wide range of applications. Moreover, to better present our work, Table 2 summarizes the key technical specifications of the hand tracking systems.

## 4. Sensor Fusion Solution

### 4.1. Extended Kalman Filter

The employed sensor fusion approach is based on the Extended Kalman Filter (EKF) framework. To estimate the hand movement, the state of the system *x* is defined as the position and velocity of each joint in the hand, which are observed through different sensors.

The state update equation is given by
(1)$$x_{k}=f\left(x_{k-1},u_{k-1}\right)+w_{k-1}$$
where f(·) represents the state transition function, $u_{k-1}$ is the control input (from Myo Armband EMG readings), and $w_{k-1}$ is the process noise.

The measurement equation is
(2)$$z_{k}=h\left(x_{k}\right)+v_{k}$$
where h(·) is the measurement function, which maps the true state space into the observed space, $z_{k}$ is the combined observation from the Leap Motion and Realsense sensors, and $v_{k}$ is the observation noise.

The prediction step in the EKF algorithm is as follows:$$\begin{array}{cc} {\hat{x}}_{k|k-1} & =f({\hat{x}}_{k-1|k-1},u_{k-1})\\ P_{k|k-1} & =F_{k}P_{k-1|k-1}F_{k}^{T}+Q_{k} \end{array}$$
where $F_{k}=\frac{\partial f}{\partial x}{|}_{{\hat{x}}_{k-1|k-1},u_{k-1}}$ is the Jacobian matrix of partial derivatives of the function *f*, $P_{k|k-1}$ is the a priori estimate error covariance, and $Q_{k}$ is the process noise covariance.

The update step in the EKF algorithm is
$$\begin{array}{cc} K_{k} & =P_{k|k-1}H_{k}^{T}{\left(H_{k}P_{k|k-1}H_{k}^{T}+R_{k}\right)}^{-1}\\ {\hat{x}}_{k|k} & ={\hat{x}}_{k|k-1}+K_{k}\left(z_{k}-h\left({\hat{x}}_{k|k-1}\right)\right)\\ P_{k|k} & =\left(I-K_{k}H_{k}\right)P_{k|k-1} \end{array}$$
where $K_{k}$ is the Kalman gain, $H_{k}=\frac{\partial h}{\partial x}{|}_{{\hat{x}}_{k|k-1}}$ is the Jacobian matrix of partial derivatives of the function *h*, $z_{k}$ is the observation from sensors, and $R_{k}$ is the observation noise covariance.

The EKF-based sensor fusion algorithm offers a systematic approach to combining information from various sensor modalities. By considering both the sensor measurements and the dynamic model, it provides an estimation of the most likely hand movements. This fusion of sensor data enables us to obtain a more accurate and comprehensive understanding of the hand’s motion.

### 4.2. LSTM Optimization

We use a Long Short-Term Memory (LSTM) network within the EKF framework to model the system dynamics. The LSTM model captures temporal dependencies in the sensor data, providing a precise representation of the hand movements.

The LSTM is a type of RNN that includes a memory cell $c_{t}$ and three gating units—the input gate $i_{t}$, the forget gate $f_{t}$, and the output gate $o_{t}$. They are updated as follows:(3)$$\begin{array}{cc} i_{t} & =\sigma (W_{ui}u_{t}+W_{hi}h_{t-1}+W_{ci}c_{t-1}+b_{i}) \end{array}$$(4)$$\begin{array}{cc} f_{t} & =\sigma (W_{uf}u_{t}+W_{hf}h_{t-1}+W_{cf}c_{t-1}+b_{f}) \end{array}$$(5)$$\begin{array}{cc} c_{t} & =f_{t}⊙c_{t-1}+i_{t}⊙\tanh\left(W_{uc}u_{t}+W_{hc}h_{t-1}+b_{c}\right) \end{array}$$(6)$$\begin{array}{cc} o_{t} & =\sigma (W_{uo}u_{t}+W_{ho}h_{t-1}+W_{co}c_{t}+b_{o}) \end{array}$$(7)$$\begin{array}{cc} h_{t} & =o_{t}⊙\tanh\left(c_{t}\right) \end{array}$$
where $W_{xy}$ are weight matrices, $b_{y}$ are bias vectors, σ(·) is the sigmoid function, tanh(·) is the hyperbolic tangent function, and ⊙ represents element-wise multiplication.

The updated state vector is defined as $x^{\prime }={\left[x^{T},h^{T},c^{T},\theta^{T}\right]}^{T}$, where *x* denotes the previous state, *h* and *c* are the hidden state and cell state of the LSTM, respectively, and $\theta =\{W_{ui},W_{hi},W_{ci},b_{i},W_{uf},W_{hf},W_{cf},b_{f},W_{uc},W_{hc},b_{c},W_{uo},W_{ho},W_{co},b_{o}\}$ are the parameters of the LSTM. The state update equation becomes
(8)$$x_{k}^{\prime }=f\left(x_{k-1}^{\prime },u_{k-1},\theta_{k-1}\right)+w_{k-1}$$
where f(·) is the function represented by the LSTM.

The convergence of this system depends on two aspects: the convergence of the EKF algorithm and the convergence of the LSTM. Provided that the LSTM can accurately model the system, the error covariance $P_{k}$ of the EKF will converge to a steady-state value.

The convergence of the LSTM depends on the chosen training algorithm and the quality of the data. Given a suitable learning rate and optimization algorithm (like Adam or RMSProp), and under the assumption that the system dynamics can be modeled by an LSTM, the weights θ will converge to a set of values that minimize the prediction error.

## 5. Performance Validation

In our hand tracking system, we employed various methods, each with its specific parameters.

The LSTM model utilized in our system has a hidden layer size of 256 and a dropout rate of 0.2. It was trained using the Adam optimizer with a learning rate of 0.001 over 50 epochs. This model handles the sensor fusion part, where data from Leap Motion, Realsense camera, and Myo Armband are combined. Its structure and training parameters are optimized to capture complex temporal dependencies in hand movement data, ensuring high real-time performance and accuracy.

The Leap Motion controllers operate at a rate of 120 frames per second, providing real-time positional and rotational information for each finger joint. The accuracy of these readings is greatly influenced by the physical placement of the Leap Motion devices. In our setup, we utilized three devices per hand to mitigate potential occlusion issues.

The Realsense depth cameras capture depth images at a resolution of 1280 × 720 pixels and 30 frames per second. These cameras provide valuable additional information about the relative depth of different parts of the hand, enhancing tracking accuracy, particularly in more complex gestures.

The Myo Armbands are configured to capture electromyography (EMG) signals at a sampling frequency of 200 Hz. These signals offer an additional layer of detail to the hand motion by monitoring the electrical activity of the forearm muscles.

In contrast, other methods, such as System A, B, C, and D, do not offer the same level of detail or temporal resolution. For example, System A employs a simpler time-delay neural network model with a hidden layer size of 128, which, although faster to train, does not capture temporal dependencies as effectively as the LSTM model. System B relies solely on Leap Motion data and thus suffers from occlusion issues in more complex hand gestures. System C and D do not incorporate depth or EMG information, resulting in less detailed hand movement tracking.

To ensure a fair comparison across different algorithms, we used the same dataset obtained from all sensors as the input for all methods. The performance, accuracy, and robustness of each method were evaluated based on the following metrics:

The real-time performance of a method was determined by its ability to process the input data and provide results within a set timeframe. This was computed as the percentage of data samples processed within this timeframe out of the total number of samples. Accuracy was determined based on how well a method’s output matched the ground truth. Specifically, for each sample, we compared the predicted hand pose with the actual (ground truth) pose. The accuracy was then calculated as follows:(9)$$\mathrm{Accuracy}=\frac{n_{\mathrm{correct}}}{N}\times 100\%$$
where *N* is the total number of samples, and $n_{\mathrm{correct}}$ is the number of samples for which the predicted hand pose matched the actual pose within a predefined error margin.

Robustness refers to the ability of a method to maintain its performance in the face of noise or disturbances in the input data. To evaluate robustness, we introduced a small random disturbance to each input data sample and then measured how well the method’s output matched the ground truth. The robustness was calculated as follows:(10)$$\mathrm{Robustness}=\frac{n_{\mathrm{robust}}}{N}\times 100\%$$
where $n_{\mathrm{robust}}$ is the number of samples for which the method’s output matched the ground truth within a predefined error margin, even when a small disturbance less than a threshold value ϵ was introduced to the input data.

As shown in Table 3, our LSTM-based system demonstrates superior performance in real-time responsiveness, accuracy, and robustness when compared to other methods like Artificial Neural Networks (System A), Support Vector Machines (System B), Random Forest (System C), and Naive Bayes (System D). This comparison underscores the effectiveness of our approach in hand-motion tracking applications.

Figure 5 provides a visual representation of the performance comparison between our system, leveraging a Long Short-Term Memory (LSTM) network, and two other systems (System A and System B) that do not employ LSTM. The metrics for comparison include real-time performance, accuracy, and robustness, reflecting the critical aspects of our application.

Our system demonstrates superior performance compared to both System A and System B across all aspects evaluated. In terms of real-time performance, our system achieves an impressive effectiveness rate of 98.9%. This high level of responsiveness ensures that actions based on hand movement recognition can be processed and responded to instantly, enhancing the overall user experience in interactive applications.

Furthermore, our system achieves an accuracy rate of 95.6%, significantly surpassing the performance of the other two systems. This highlights the potential of LSTM networks in recognizing complex, non-linear patterns in temporal data, thus enabling highly accurate hand motion tracking.

The robustness of our system is also noteworthy. The LSTM-based system demonstrates exceptional resilience in the face of noise, disturbances, and changes in the operating environment. This underscores its suitability for reliable and practical deployment in real-world settings.

In summary, the results presented in the table strongly support the superiority of our LSTM-based hand motion tracking system in terms of real-time performance, accuracy, and robustness. This reaffirms the selection of LSTM as the foundational technology for our system.

## 6. Conclusions and Future Work

Our novel sensor fusion approach, combined with LSTM-based data processing, demonstrates significant advancements in hand tracking accuracy and robustness for VR-based HCI applications. By harnessing a multitude of sensors, we achieve comprehensive and complementary data capture, enabling the precise real-time tracking of complex hand movements.

Future work can explore the integration of additional sensing modalities and improved algorithms for data fusion to enhance the robustness and accuracy of the system further. As VR technologies continue to evolve, the continuous refinement of sensor fusion techniques and hand tracking algorithms is crucial for keeping pace with advancements in the field. While our system has demonstrated impressive performance in a controlled environment, its effectiveness in real-world applications and user acceptance remains to be validated in future studies.

Furthermore, in this paper, a wall was implemented between the two interaction cubes in this study to avoid cross-interference from simultaneous hand movements. We acknowledge that this design choice might limit certain interaction scenarios where both hands need to cooperate, such as one hand assisting the other, which is a frequent situation in real-world tasks. While the current setup was designed this way to ensure precise data collection and analysis in our initial explorations, we acknowledge this as a limitation of the current work. We foresee future modifications of our system setup to better accommodate natural dual-hand interactions. Our future work will focus on eliminating the physical barrier and instead employ advanced algorithms capable of distinguishing and tracking individual hand movements in a shared space, thus facilitating more naturalistic interaction scenarios.

## Figures and Tables

**Figure 1 biomimetics-08-00326-f001:**
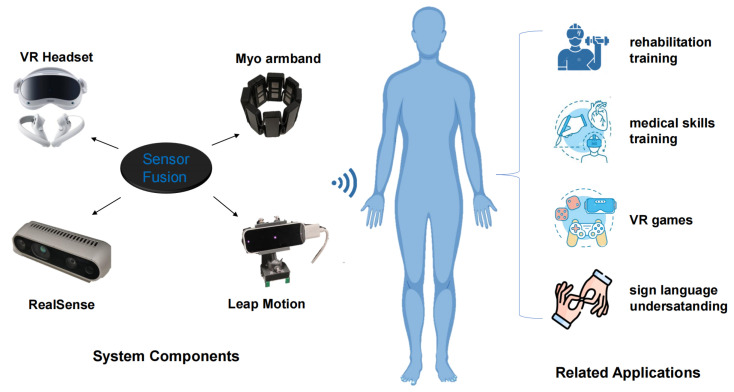
The proposed system and related application scenarios.

**Figure 2 biomimetics-08-00326-f002:**
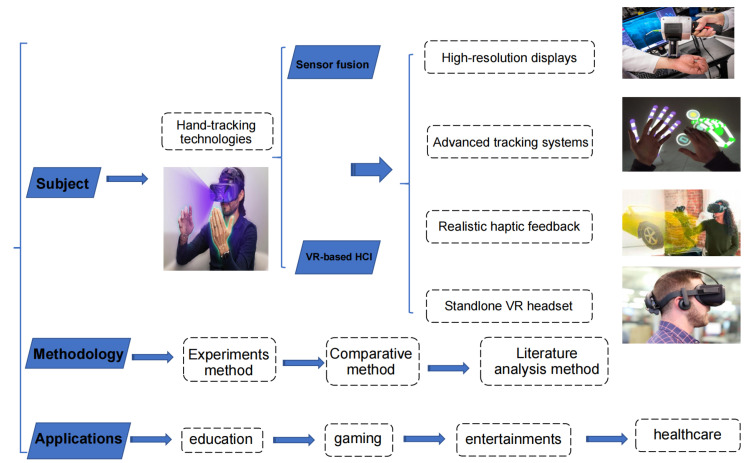
The integration overview framework.

**Figure 3 biomimetics-08-00326-f003:**
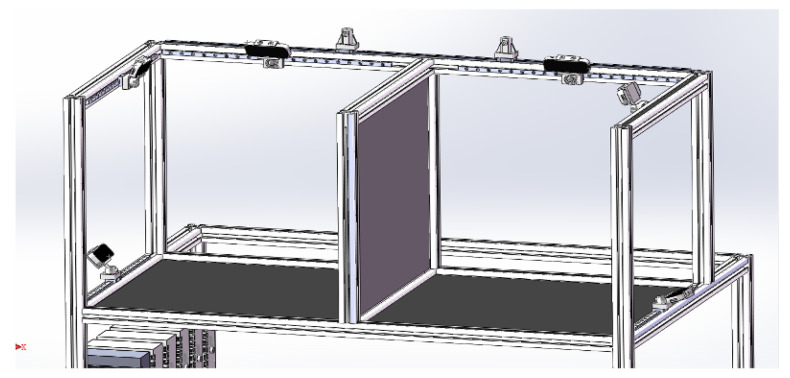
The developed hand tracking device.

**Figure 4 biomimetics-08-00326-f004:**
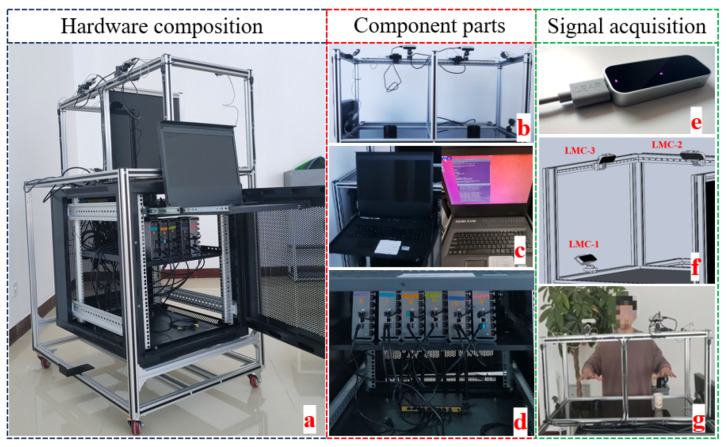
The detailed information of the developed hand tracking device. A high-level system diagram illustrating the integration of various components of our hand tracking system. For each hand, three Leap Motion controllers, a Realsense camera, and a Myo Armband are managed by a computer running ROS. Processed data are communicated to a VR environment running on a PICO 4 device. (**a**): Hardware composition; (**b**–**d**): Component parts; (**e**–**g**): Signal acquisition.

**Figure 5 biomimetics-08-00326-f005:**
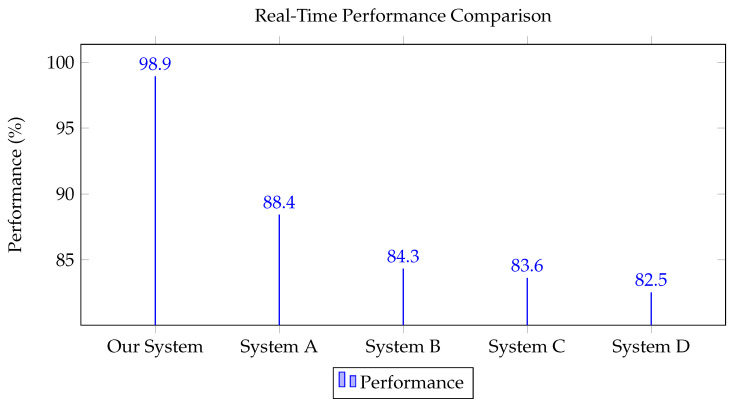
Real-time performance comparison.

**Table 1 biomimetics-08-00326-t001:** Comparison of hand-tracking systems.

Features	This System	Optical Marker-Less Hand-Tracking Systems	NUI	ADAS	HPE	Fast Mobile Eye Tracking
Sensing Modalities	EMG, Hand tracking	Hand tracking	Motion tracking, Triangulation	Lidar, Camera, Radar	RGB, RF, Radio frequency-tracking	IR, Eye tracking
Stereo Vision	Yes	No	No	No	Yes	Yes
Capture Depth Information of A Large Area	Yes	No	No	Yes	No	No
Inside-out Tracking	Yes	Yes	No	Yes	Yes	No
3D Environment	Yes	No	Yes	No, 2D lidar	Yes	No
Research Grade	Yes	Yes	Yes	Yes	Yes	Yes

**Table 2 biomimetics-08-00326-t002:** Key technical specifications of the hand tracking systems.

Technical Specifications	Details
Hand Tracking Method	Multi-sensor fusion incorporating Leap Motion, Realsense depth camera, and Myo Armband
Number of Sensors per Hand	5 (3 Leap Motion controllers, 1 Realsense camera, 1 Myo Armband)
Computer System	Ubuntu 20.04 LTS running ROS Noetic Ninjemys
Data Processing	MATLAB 2023a
Virtual Reality Environment	Unity 2023.1.0, running on PICO 4
Number of Joints Tracked	22 minor joints per hand
Depth Perception	Provided by Realsense depth camera
EMG Data Collection	Myo Armband
Tracking Accuracy	0.7 mm (typical for Leap Motion)
Depth Image Resolution	1280 × 720 (typical for Realsense camera)
EMG Sampling Rate	200 Hz (typical for Myo Armband)
Latency	10 ms (average for sensor data fusion systems)

**Table 3 biomimetics-08-00326-t003:** Comparison of the performance of different systems/algorithms for hand motion tracking.

System/Algorithm	Real-Time Performance (%)	Accuracy (%)	Robustness (%)
Our System (LSTM based)	98.9	95.6	94.3
System A (ANN based)	88.4	85.7	80.3
System B (SVM based)	84.3	78.2	75.5
System C (Random Forest based)	83.6	77.4	70.3
System D (Naive Bayes based)	82.5	75.3	65.7

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
