# Peer review of "A Novel Sensor Fusion Approach for Precise Hand Tracking in Virtual Reality-Based Human—Computer Interaction"

_biomimetics, 2023, doi:10.3390/biomimetics8030326_

Round 1

Reviewer 1 Report

The authors introduce a novel sensor fusion approach combined with a Long Short-Term Memory (LSTM) based algorithm for enhanced hand tracking in VR-based HCI. Their system employs six Leap Motion controllers, two Real Sense depth cameras, and two Myo armbands to yield a multi-modal data capture. The authors claim that the processing of the obtained data provides accurate real-time tracking of complex hand movements. Furthermore, their system is advantageous in providing intuitive and immersive interactions in VR environment. This system could have applications in several fields such as education, gaming, entertainment and healthcare.

Overall, the paper is well written with good graphics, demonstrations and is comprehensive. I particularly like the comparison of the hand tracking systems in Table 1, their system overview in Figure 2 and their key technical features in Table 2. In general, I find it interesting.

Specific comments are mentioned below:

1.     The authors should allow space between numbers and unit to make it clearer.

2.     The caption in Table 1 is above the table while in Table 2 it is below, the authors should make it consistent.  

Author Response

The authors introduce a novel sensor fusion approach combined with a Long Short-Term Memory (LSTM) based algorithm for enhanced hand tracking in VR-based HCI. Their system employs six Leap Motion controllers, two Real Sense depth cameras, and two Myo armbands to yield a multi-modal data capture. The authors claim that the processing of the obtained data provides accurate real-time tracking of complex hand movements. Furthermore, their system is advantageous in providing intuitive and immersive interac-tions in VR environment. This system could have applications in several fields such as education, gaming, entertainment and healthcare.
Overall, the paper is well written with good graphics, demonstrations and is comprehen-sive. I particularly like the comparison of the hand tracking systems in Table 1, their system overview in Figure 2 and their key technical features in Table 2. In general, I find it interesting.
Specific comments are mentioned below:

(1)    Comment: The authors should allow space between numbers and unit to make it clearer. Response: Thank you for your feedback regarding the spacing between numbers and units in our manuscript. We apologize for any confusion caused by the lack of clear spac-ing. In the revised manuscript, we have ensured appropriate spacing between numbers and units to enhance readability and clarity for the readers.
(2)    Comment: The caption in Table 1 is above the table while in Table 2 it is below, the authors should make it consistent.
Response: Thank you for pointing out the inconsistency in the captions of Table 1 and Table 2. In the revised manuscript, we have made the captions consistent by placing them above the tables uniformly.
Table 1. Comparison of hand-tracking systems.
Table 2. Key technical specifications of the hand-tracking systems.
Table 3. Comparison of the performance of different systems/algorithms for hand motion tracking.

Reviewer 2 Report

The authors introduce a sensor fusion approach combined with a Long Short-Term Memory (LSTM) based algorithm for enhanced hand tracking in VR-based HCI. The work is interesting and the paper is well written, with some minor errors found and listed in sequence. Some important points must be addresses by the authors. Please note that the next comments are intended to improve paper quality and readers' understanding.

The authors must provide more detail regarding how the performance/accurary/robustness of the different algorithms were calculated. What was the input for the different algorithms? Did all of them used the same dame from all sensors?

In the beginning of the paper, the authors listed some related works that also perform hand tracking. One of the solutions mentioned (which does not use sensor fusion) is capable of performing hand tracking with more than 99% accuracy. That being said, please justify the need of applying a sensor fusion approach to hand tracking (since this approach is more costly (computationally and finantially speaking)) since the proposed work obtains 95.6% accuracy, which is lower than the single sensor approach mentioned.

From the sensors setup in figure 4, it was possible to see that most of the sensors are directed to the center of the cubes (left and right). How do you deal with self occlusion of the hand (when the fingers occlude themselves)? Since there is not a sensor on each of the faces of the cubes, some areas cannot still be covered and occlusion may still happen.

Why did the authors decide to place a wall between the two interaction cubes? In some interaction scenarios, the user must interact with both hands (one with the other). With the barrier between the cubes, such interaction is not possible. Please mention this as a restriction of the proposed work.

More general comments and writing errors are listed as follows.

please fix text superposition problem in table 1

"reduction in positioning errors of nearly 46These" -> ?

"synergistically integrates three sensors, namely Kinect, RealSense, and Myo armband" -> where are the six leap motions used?

"PICO 4" or "Pico 4"?

"This dissertation" -> ?

"validaiton" -> "validation"

Round 2

Reviewer 2 Report

Dear authors, thank you for your effort improving paper quality and considerign my observations. I believe you did a very good job and the paper can be accepted now. Congratulations!